# Tuberculosis among Ambulatory People Living with HIV in Guangxi Province, China: A Longitudinal Study

**DOI:** 10.3390/ijerph191912280

**Published:** 2022-09-27

**Authors:** Zhezhe Cui, Fei Huang, Dabin Liang, Yan Huang, Huifang Qin, Jing Ye, Liwen Huang, Chongxing Zhou, Minying Huang, Xiaoyan Liang, Fengxue Long, Yanlin Zhao, Mei Lin

**Affiliations:** 1Guangxi Key Laboratory of Major Infectious Disease Prevention and Control and Biosafety Emergency Response, Guangxi Zhuang Autonomous Region Center for Disease Control and Prevention, Nanning 530028, China; 2National Center for TB Control and Prevention, China CDC, Beijing 100013, China; 3School of Public Health, Guangxi Medical University, Nanning 530021, China

**Keywords:** tuberculosis, HIV/AIDS, ambulatory, China, coinfection

## Abstract

Background: This study aims to determine the prevalence of TB among ambulatory people living with HIV in Guangxi Province, which experienced the biggest HIV epidemic in China. Methods: We undertook a longitudinal study in five HIV/AIDS designated hospitals randomly selected from Guangxi Province; all newly diagnosed HIV/AIDS outpatients from 2019 to 2021 were screened for TB and interviewed with a questionnaire. Results: A total of 4539 HIV/AIDS outpatients were enrolled, with 2886 (63.6%) men and 1653 (26.4%) women. The prevalence of TB/HIV coinfection was 0.8%, with a clear downward trend from 1.3% in 2019 to 0.4% in 2021 (*p* = 0.0011). The prevalence of LTBI was 24.3%, with no significant differences from 2019 to 2021. The percentages of AIDS, comorbidity, nine symptoms and abnormal chest X-ray of TB were higher than those of the other PLWH. Conclusion: The prevalence of TB among ambulatory people with HIV in Guangxi Province was 14 times higher than the general population, and the annual declined TB prevalence indicated the effectiveness of TB and HIV control and prevention over recent years. The findings proved that symptom screening was insufficient for TB diagnosis and highlighted the importance of systematic TB screening at every visit to a health facility.

## 1. Introduction

Tuberculosis (TB) and human immunodeficiency virus/Acquired immunodeficiency syndrome (HIV/AIDS) have been considered as the two of top three infectious diseases worldwide in terms of disease burden, which are still threatening humans in the modern era since their origin [1]. Unlike other opportunistic infections, TB disproportionately affects people living with HIV even before any significant drop in CD4+ T cell counts [2]. While TB and HIV coinfection provides reciprocal advantages to both pathogens, including HIV infection increasing the risk of active TB and TB infection increasing the level of HIV replication, coinfection has fueled the TB epidemic, presenting programmatic management and treatment challenges across the world [3]. According to a global TB report of the World Health Organization (WHO), people living with HIV are 18 times more likely to develop active TB disease than people without HIV, and there were about 787,000 TB/HIV-coinfected patients, which accounted for about 7.9% of the total new TB patients and 204,000 TB deaths among people living with HIV (PLWH) in 2020 globally [4].

As the country with the second highest TB burden after India, China has placed great emphasis on infectious disease control and achieved substantial progress both in TB and HIV over the recent decades [5,6]. However, TB and HIV remain two major public health issues in this country. In 2020, there were an estimated 842,000 incident TB cases in China, of which 12,000 were TB/HIV-coinfected patients and 2100 were TB deaths among people living with HIV [4]. However, the epidemic of TB and HIV varies across China. Guangxi Zhuang autonomous region (Guangxi Province), located in southwestern China, is has the highest HIV prevalence in China [7], where HIV/TB coinfection is one of the most common coinfections [8].

To minimize HIV-associated TB burden, early diagnosis and symptomatic screening are necessary and crucial, which could improve the diagnosis delay and stop the probability of transmission and continuation of epidemic and are in line with the priority of the WHO’s End TB Strategy [9]. This study aimed to explore the prevalence and risk factors of TB/HIV coinfection in Guangxi Province, which could be referred by policymakers to address the more vulnerable population and formulate a more comprehensive screening strategy.

## 2. Materials and Methods

### 2.1. Study Design and Participants

We carried out a longitudinal study in prefecture-level HIV/AIDS designated hospitals in five cities, which were randomly selected from Guangxi Province from 2019 to 2021. The five cities were Nanning, Liuzhou, Laibin, Chongzuo, and Guigang, which make up about 20 million people in terms of population (Figure 1). All newly diagnosed HIV/AIDS outpatients without known TB history at the prefecture-level HIV/AIDS designated hospitals of the five cities were enrolled in this study.

### 2.2. Procedures

All eligible HIV/AIDS outpatients were diagnosed in the five HIV/AIDS designated hospitals using chest X-ray, interferon γ release assays (IGRAs), three sputa (morning, spot, and evening) for smear microscopy and culture on solid Löwenstein–Jensen media. Ultrasonic atomization-induced sputum collection was recommended for patients with no cough or low sputum volume. The three sputa were delivered to the provincial tuberculosis reference laboratory of Guangxi Center for Disease Control and Prevention (Guangxi CDC) for smear and culture double checking. The quality of provincial tuberculosis reference laboratories was ensured and evaluated annually by the national reference laboratory of the Chinese Center for Disease Control and Prevention (China CDC).

After obtaining informed consent, trained physicians interviewed all enrolled participants with a questionnaire on demographic information, comorbidities and symptoms, and complemented the following examination results, including CD4 cell count, chest X-ray image evaluation, sputum smear and culture, and IGRAs. There were 16 comorbidities and 9 symptoms in the questionnaire. The nine symptoms included cough, fever, night sweats, weight loss, hemoptysis, chest pain, loss of appetite, shortness of breath, and swollen lymph nodes. The questionnaire was then double-checked by another trained physician. The questionnaires were shipped to Guangxi CDC and randomly selected for rechecking again by the quality controller before entering into an electronic database in parallel.

### 2.3. Statistical Analysis

We compared demographics and clinical characteristics among TB, latent tuberculosis infection (LTBI) and HIV mono-infection among PLWH. In terms of the symptoms, we compared the difference of the WHO four-symptom screen (W4SS: comprising current cough, fever, night sweats, or weight loss) in addition to the nine symptoms.

Quantitative data are expressed as the median and interquartile range (IQR), and categorical variables are expressed as the number and percentage. Wilcoxon Scores (Rank Sums) were used for quantitative data. The χ^2^ (Chi-square) test was used for categorical data, and Fisher’s exact test was used when the χ^2^ test was not applicable. All tests were two-tailed, and the level of statistical significance was set at *p* < 0.05.

### 2.4. Ethical Consideration

The study protocol was reviewed and approved by the Ethical Committee of Guangxi CDC. Informed consent was obtained from all participants.

## 3. Results

### 3.1. TB Diagnosis

A total of 4539 outpatients were enrolled, and the numbers of newly diagnosed HIV/AIDS outpatient in 2019, 2020 and 2021 were 1865, 675, and 1999, respectively, with a sharp decline in 2020 and a quick rebound in 2021. In total, there were 36 TB cases, 66 Nontuberculous Mycobacterial (NTM) cases and 1104 LTBI cases were detected, and the other 3333 cases were HIV mono-infection and did not get infected with mycobacterium. Extra-pulmonary tuberculosis (EPTB) accounted for 11.1% of all TB cases, while NTM cases comprised 64.7% of all cases isolated with mycobacterium. The prevalence of TB infection among ambulatory PLWH was 0.8%, with a clear downward trend from 1.3% in 2019 to 0.4% in 2021 (*p* = 0.0005). However, the prevalence of LTBI among ambulatory PLWH was 24.3%, with no significant differences from 2019 to 2021 (Table 1).

### 3.2. Demographics

Among the 4539 ambulatory PLWH, 2886 (63.6%) were male and 1653 (26.4%) were female, with significant difference among TB, LTBI and HIV mono-infection. The majority of PLWH were married, with primary and junior high school education, and worked as farmers. However, for all of them, except for occupation among TB cases, there were no significant differences among TB, LTBI and HIV mono-infection (Table 2). The median age (IQR) was 48 (39–59). The main age groups were the 45 group, 35 group and 55 group, which accounted for 26.2%, 25.1% and 17.5% of the total cases, respectively (Figure 2 and Appendix A Table A1).

### 3.3. Clinical Characteristics

Different clinical characteristics between active TB, LTBI and HIV mono-infection were compared. There were no significant differences for all clinical characteristics among LTBI and HIV mono-infection. There were no significant differences regarding the percentages of close contact and none for the four symptoms among TB/HIV coinfection patients as well. However, the percentages of AIDS, comorbidity, nine symptoms and chest X-ray of TB were higher than those of the other PLWH (Table 3).

The median (IQR) CD4 cell counts of TB, LTBI, and HIV mono-infection were 241 (183–458), 425 (268–606) and 387 (236–568), respectively. The CD4 cell count of HIV cases was higher than that of AIDS cases in both groups of LTBI (*p* < 0.0001) and HIV mono-infection (*p* < 0.0001), while there was no difference in the TB/HIV coinfection group (*p* = 0.6077) (Figure 3 and Appendix A Table A2).

## 4. Discussion

Tuberculosis is a very common opportunistic infection in people living with HIV. A previous study in China reported that the prevalence of TB/HIV coinfection among PLWH ranged from 4.8 to 15.7% [10,11,12]. However, our study revealed that the prevalence of TB among ambulatory PLWH was 0.8%, which is much lower than this previous study. The main possible reason for this phenomenon is that our study only enrolled newly diagnosed ambulatory HIV-infected individuals. These patients were not hospitalized as they were in good physical condition, which indicates that their immune system may not be greatly suppressed by HIV, and the chance for them to develop active TB was not as high as supposed. The possible reason for the annually declining prevalence of TB was due to the increased awareness of HIV active testing in China; more and more people will go to voluntary counseling and testing (VCT) clinics once they exhibit high-risk behaviors, and the chance for them to develop TB is low as they are at the very early stage of HIV infection with normal immune function. Nevertheless, the overall prevalence of TB among ambulatory PLWH was about 14 times higher than the prevalence of 59 new incident TB cases per 100,000 of the general population in China [4]. NTM accounted for about two-thirds of all cases isolated with mycobacterium, 10 times higher than the 6.4% previously reported in China [13]. The possible reason for this is immunosuppression. In addition, the prevalence of LTBI among ambulatory PLWH was 24.3%, which is higher than the prevalence of 19.8% reported in rural area of China and 5.8% reported in the USA [14,15]. Infection with HIV is the most powerful known risk factor predisposing one to mycobacterium tuberculosis infection and progression to active disease, which increases the risk of latent TB reactivation 20-fold [16]. Such a reactivation of tuberculosis can be averted by tuberculosis-preventive treatment (TPT) [17]. TPT is highly effective, especially in people living with infection, in whom it reduces the risk of death by a third, with or without concurrent antiretroviral therapy (ART) [18]. It is necessary for governments to pay more attention to and allocate more resources for the two diseases.

The World Health Organization recommended that PLWH be routinely screened for tuberculosis with W4SS over the past decade [19]. If W4SS is positive, the patient should then receive a molecular, WHO-recommended rapid diagnostic test [20]. However, our study showed that the percentages of W4SS were no different among TB, LTBI, and HIV mono-infection groups, while the percentages of nine symptoms among TB patients were higher than that among LTBI and HIV mono-infection. These results indicate that W4SS may not be a good option for tuberculosis screening in people living with HIV and more symptoms should be considered. Half of the TB cases in our study presented abnormal Chest X-rays, more than twice than those of LTBI and HIV mono-infection. The findings from another study confirmed that W4SS had suboptimal sensitivity and chest X-ray could be used in parallel with W4SS because this strategy could detect more tuberculosis cases than W4SS alone [21]. The latest WHO consolidated guidelines on tuberculosis recommend that PLWH should be screened systematically for TB at every visit to a healthcare facility, which is essential to reduce TB-related morbidity and mortality [22].

The risk of TB/HIV occurrence was found to be high among PLWH in WHO clinical stage III (AIDS) [23]. The prevalence of TB in AIDS in our study was higher than that in HIV, while the prevalence of LTBI had no difference between AIDS and HIV. As TB and HIV act in synergy, TB risk increases in PLWH as immune functions declined and CD4 T cell count decreases. The odds of getting TB in PLWH was 1.43 times higher for every 100/μL decrease in CD4 cell count. A CD4 cell count of less than 200/μL was one of the possible risk factors elucidated for TB/HIV coinfection [24]. Our study also observed that the CD4 cell count of TB/HIV coinfection cases was lower than that of both LTBI and HIV mono-infection patients among PLWH. However, declining TB rates in PLWH were observed over time and with CD4 recovery, highlighting the importance of early and successful antiretroviral therapy (ART) for disease control and prevention [25].

The COVID-19 pandemic has had a devastating impact on the fight against TB, HIV and other deadly infectious diseases, according to a report that compares 2019 and 2020 data in more than 100 low- and lower-middle-income countries [26]. China experienced a sharp decline of TB notification after the COVID-19 pandemic and it took about 10 weeks to get back to the normal level of TB notification in 2020 [27]. The PLWH notification from the present study showed a similar pattern, with an obvious decrease in 2020 and a quick rebound in 2021. This further confirmed the impact of COVID-19 on HIV control and prevention in China, which meant that a lot of PLWH did not get diagnosed and treated and in turn increased the likelihood of community transmission and the chance of developing active TB. In terms of the demographics of notified PLWH, male cases and the age group of 35–64 years old accounted for about two-thirds of the total cases, which is in line with a previous study on Guangxi Province [28]. Health education and intervention on the vulnerable population, including peer education and preventive intervention for pre-exposure HIV, should be promoted for equitable and accessible health.

Although the high prevalence of TB among PLWH reemphasizes the importance of TB screening among PLWH, the diagnosis of TB among PLWH is challenging. The currently increasing rates of HIV/AIDS-associated tuberculosis have shifted the clinical pattern of TB towards smear negative pulmonary TB (PTB) and extra-pulmonary TB (EPTB), which in turn causes difficulties in the diagnosis and treatment of TB due to an unusual clinical picture with increased smear negative acid fast bacilli PTB, atypical findings upon chest radiography and increased prevalence of EPTB [29,30]. At present, sputum remains the mainstay of TB diagnostic testing and is more difficult to obtain in adults with TB/HIV coinfection, due to paucibacillary disease and the lack of cavitation, which are associated with a higher rate of smear negative disease [31]. In a meta-analysis of autopsy studies among people living with HIV, almost 50% of tuberculosis-related deaths were undiagnosed at the time of death [32]. The WHO estimates that 44% of TB disease goes undiagnosed in PLWH [33]. However, CD4+ lymphocyte count and viral load may be considered as valuable predictors for TB development [34]. Rapid ART initiation and appropriate TPT can be potential key interventions to tackle the TB epidemic and should be taken immediately to reduce mortality among PLWH in TB/HIV high-burden settings [35].

## 5. Conclusions

The prevalence of TB among ambulatory people with HIV in Guangxi Province was 14 times higher than that of the general population, and the annual declining TB prevalence indicated the effectiveness of TB and HIV control and prevention over recent years. The findings obtained in this study are important for both service providers and patients, which indicates that symptom screening is not sufficient for TB detection and highlights the importance of systematic TB screening among PLWH at every visit to a health facility, as well as mass media interventions for promoting HIV testing once after the high-risk behavior.

## 6. Limitation

Our study is subject to several limitations. Firstly, all patients were enrolled in the clinics with limited examination, which may lead to a missed EPTB diagnosis in our study. Secondly, the follow-up results and treatment outcome were not analyzed to compare PLWH with/without TB.

## Figures and Tables

**Figure 1 ijerph-19-12280-f001:**
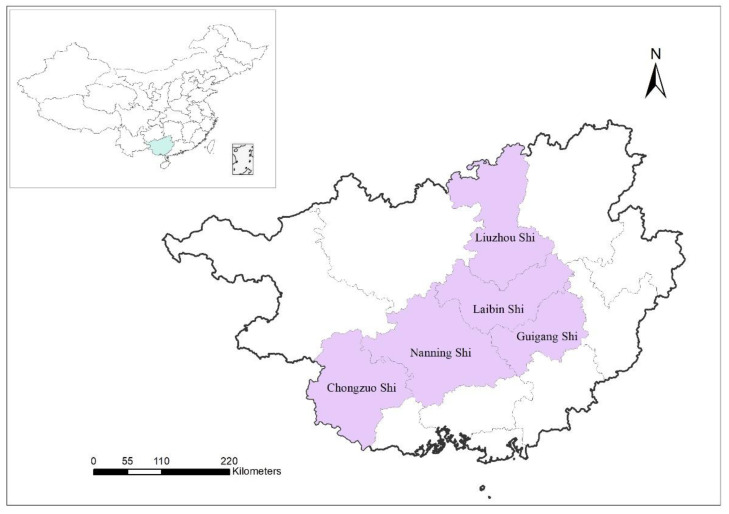
Geographical location of the five cities in Guangxi Province.

**Figure 2 ijerph-19-12280-f002:**
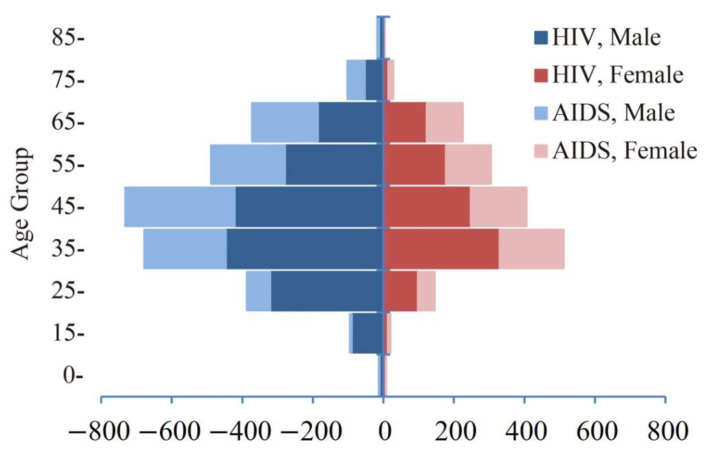
Age-sex pyramid of HIV/AIDS outpatients of Guangxi Province.

**Figure 3 ijerph-19-12280-f003:**
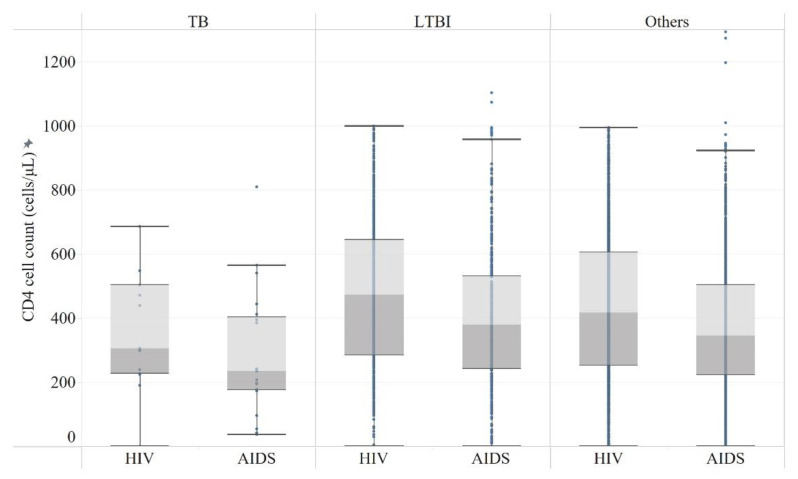
CD4 cell count of HIV/AIDS outpatients in Guangxi Province.

**Table 1 ijerph-19-12280-t001:** TB diagnosis of HIV/AIDS outpatients in Guangxi Province, 2019–2021.

Year	Total	Active TB				NTM		LTBI	
		PTB (%)	EPTB (%)	Total (%)	*p* Value *	Total (%)	*p* Value	Total (%)	*p* Value
2019	1865	22 (88.0)	3 (12.0)	25 (1.3)	0.0005	43 (2.3)	<0.0001	426 (22.8)	0.0540
2020	675	4 (100.0)	0 (0.0)	4 (0.6)		16 (2.4)		185 (27.4)	
2021	1999	6 (85.7)	1 (14.3)	7 (0.4)		7 (0.4)		493 (24.7)	
Total	4539	32 (88.9)	4 (11.1)	36 (0.8)		66 (1.5)		1104 (24.3)	

Note: PTB: pulmonary tuberculosis. EPTB: extra-pulmonary tuberculosis. NTM: nontuberculous mycobacteria. LTBI: latent tuberculosis infection. *: Cochran–Armitage trend tests.

**Table 2 ijerph-19-12280-t002:** Demographics of HIV/AIDS outpatients in Guangxi Province.

Factors	Total	Active TB		LTBI		HIV Mono-Infection	
		*n*	*p* Value	*n* (%)	*p* Value	*n* (%)	*p* Value
**Gender**			0.0336		0.0256		0.0082
Male	2886 (63.6)	29 (80.6)		733 (66.4)		2124 (62.5)	
Female	1653 (26.4)	7 (19.4)		371 (33.6)		1275 (37.5)	
**Age**			0.9212		0.8918		0.8388
0-	21 (0.5)	0 (0.0)		4 (0.4)		17 (0.5)	
15-	1837 (40.5)	13 (36.1)		440 (39.9)		1384 (40.7)	
45-	1608 (35.4)	14 (38.9)		398 (36.1)		1196 (35.2)	
60-	1073 (23.6)	9 (25.0)		262 (23.7)		802 (23.6)	
**Marriage**			0.7046		0.4471		0.4045
Married	2857 (62.9)	20 (55.6)		694 (62.9)		2143 (63.0)	
Unmarried	996 (21.9)	10 (27.8)		234 (21.2)		752 (22.1)	
Divorced/widowed	641 (14.1)	6 (16.7)		168 (15.2)		467 (13.7)	
Unknown	45 (1.0)	0 (0.0)		8 (0.7)		37 (1.1)	
**Education**			0.3496		0.0818		0.0859
Illiteracy	321 (7.1)	3 (8.3)		77 (7.0)		241 (7.1)	
Primary school	1564 (34.5)	17 (47.2)		398 (36.1)		1149 (33.8)	
Junior high school	1597 (35.2)	12 (33.3)		351 (31.8)		1234 (36.3)	
High school	586 (12.9)	3 (8.3)		157 (14.2)		426 (12.5)	
Junior college/above	471 (10.4)	1 (2.8)		121 (11.0)		349 (10.3)	
**Occupation**			0.0260		0.4583		0.5181
Farmer	2766 (60.9)	23 (63.9)		651 (59.0)		2092 (61.5)	
Self-employed	759 (16.7)	2 (5.6)		198 (17.9)		559 (16.4)	
Unemployed	527 (11.6)	9 (25.0)		132 (12.0)		386 (11.4)	
Other	487 (10.7)	2 (5.6)		123 (11.1)		362 (10.7)	

**Table 3 ijerph-19-12280-t003:** Clinical characteristics of HIV/AIDS outpatient in Guangxi province.

Factors	Total	Active TB		LTBI		HIV Mono-Infection	
		*n* (%)	*p* Value	*n* (%)	*p* Value	*n* (%)	*p* Value
**HIV status**			0.0097		0.6446		0.9420
HIV infected	2835 (62.5)	15 (46.9)		696 (61.8)		2124 (62.8)	
AIDS	1704 (37.5)	21 (65.6)		408 (36.2)		1275 (37.7)	
**Comorbidity**			0.0347		0.6675		0.3915
Yes	421 (9.3)	7 (19.4)		106 (9.6)		308 (9.1)	
No	4118 (90.7)	29 (80.6)		998 (90.4)		3091 (90.9)	
**Close contact**			0.3555		0.7313		0.6305
Yes	157 (3.5)	2 (5.6)		40 (3.6)		115 (3.4)	
No	4382 (96.5)	34 (94.4)		1064 (96.4)		3284 (96.6)	
**Nine Symptoms**			0.0052		0.6504		0.8098
Yes	276 (6.1)	7 (19.4)		64 (5.8)		205 (6)	
No	4263 (93.9)	29 (80.6)		1040 (94.2)		3194 (94)	
**W4SS ***			0.0728		0.8812		0.8007
Yes	201 (4.4)	4 (11.1)		48 (4.3)		149 (4.4)	
No	4338 (95.6)	32 (88.9)		1056 (95.7)		3250 (95.6)	
**Chest x-ray**			<0.0001		0.4773		0.0886
Abnormal	997 (22)	20 (55.6)		251 (22.7)		726 (21.4)	
Normal	3542 (78)	16 (44.4)		853 (77.3)		2673 (78.6)	

* W4SS: the WHO four-symptom screen, comprising cough, fever, night sweats, or weight loss.

## Data Availability

The datasets used in this study are not publicly available due to a confidentiality policy, but data sharing request will be considered by the research team upon written request to the corresponding author.

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
