# Peer review of "Tuberculosis among Ambulatory People Living with HIV in Guangxi Province, China: A Longitudinal Study"

_ijerph, 2022, doi:10.3390/ijerph191912280_

Round 1
Reviewer 1 Report
In this article, the authors investigated the prevalence of tuberculosis among HIV-positive ambulatory individuals in Guangxi province, China. The authors enrolled 4539 HIV/AIDS outpatients and reported that TB/HIV co-infection was 0.8% prevalent. The authors also noted that the incidence of tuberculosis is 14 times higher among HIV-positive individuals than among those without the HIV. However, the percentage of TB/HIV co-infection has declined from 1.3% to 0.4% between 2019 and 2021. The work is significant and demonstrates the need of studying TB-HIV co-infection.
Few minor comments:
1. Line 111, expand NTM here since the authors are mentioning NTM first in this sentence.
2. Line 113, same for EPTB, expand EPTB here
3. In table 1: expand PTB
4. In table 2, please add a row with age and its statistical significance. There is a figure with age, but it does not include statistical values. Also, if the authors have the information, please add BMI also. Age and BMI are the two most important factors in the case of both TB and HIV.
5. Line 136: Please write active TB, LTBI, and HIV. "active" word is missing from TB.
Author Response
1. Line 111, expand NTM here since the authors are mentioning NTM first in this sentence.
Response: thanks for comment, we have revised as suggested.
2. Line 113, same for EPTB, expand EPTB here
Response: thanks for comment, we have revised as suggested.
3. In table 1: expand PTB
Response: thanks for comment, we have revised as suggested.
4. In table 2, please add a row with age and its statistical significance. There is a figure with age, but it does not include statistical values. Also, if the authors have the information, please add BMI also. Age and BMI are the two most important factors in the case of both TB and HIV.
Response: thanks for comment, we have added a row with age in table 2. However, we didn’t collect weight and height to get BMI in the study.
5. Line 136: Please write active TB, LTBI, and HIV. "active" word is missing from TB.
Response: thanks for comment, we have revised as suggested.
Reviewer 2 Report
The manuscript reports the cases of tuberculosis-HIV coinfection in Guangxi (China). The reported data is useful for epidemiological statistics and the manuscript is well written. There are only a few comments that may help improving the quality of the article before being accepted for publication at IJERPH.
· Introduction: the authors mentioned that tuberculosis has an opportunistic behaviour even when the levels of CD4+ haven’t dropped. Is there any explanation for this? If so, please report.
· The authors mentioned that about 787,000 patients are coinfected with TB and HIV. Have they found any percentage so the amount of coinfected patients can be compared with the total amount of HIV patients?
· Line 125: ‘farmer’ instead of ‘famer’.
· Line 233: ‘higher’ instead of ‘high’.
Author Response
Introduction: the authors mentioned that tuberculosis has an opportunistic behaviour even when the levels of CD4+ haven’t dropped. Is there any explanation for this? If so, please report.
Response: Thanks for the comment. TB is an airborne infectious disease and all people are likely to be infected with TB regardless of the level of CD4+.
The authors mentioned that about 787,000 patients are coinfected with TB and HIV. Have they found any percentage so the amount of coinfected patients can be compared with the total amount of HIV patients?
Response: thanks for the comment. There were about 9.9 million people fell ill with TB in 2020, and the number of TB/HIV coinfection accounted for about 7.9% of the new TB patients in 2020. We have added the sentence in the manuscript. However, we didn’t know whether these people coinfected with TB/HIV were newly infected with HIV.
Line 125: ‘farmer’ instead of ‘famer’.
Response: thanks for comment, we have revised as suggested.
Line 233: ‘higher’ instead of ‘high’.
Response: thanks for comment, we have revised as suggested.